# Endothelial Nitric Oxide Mediates the Anti-Atherosclerotic Action of *Torenia concolor* Lindley var. *Formosama* Yamazaki

**DOI:** 10.3390/ijms21041532

**Published:** 2020-02-24

**Authors:** Li-Ching Cheng, Bei-Chia Guo, Chia-Hui Chen, Chi-Jen Chang, Ta-Sen Yeh, Tzong-Shyuan Lee

**Affiliations:** 1Department of Nursing, Division of Basic Medical Sciences, Chang Gung University of Science and Technology, Taoyuan 33303, Taiwan; victoria@gw.cgust.edu.tw; 2Graduate Institute and Department of Physiology, College of Medicine, National Taiwan University, Taipei 10051, Taiwan; d07441002@ntu.edu.tw; 3Institute and Department of Physiology, School of Medicine, National Yang-Ming University, Taipei 11221, Taiwan; chiahui1993@gmail.com; 4The First Cardiovascular Division, Chang Gung Memorial Hospital, Taoyuan 33305, Taiwan; cchijen@adm.cgmh.org.tw; 5Department of General Surgery, Chang Gung Memorial Hospital, Chang Gung University, Taoyuan 33305, Taiwan

**Keywords:** *Torenia concolor* Lindley var. *formosama* Yamazak, endothelial nitric oxide synthase, nitric oxide, anti-inflammatory effect, atherosclerosis

## Abstract

*Torenia concolor* Lindley var. *formosama* Yamazaki ethanolic extract (TCEE) is reported to have anti-inflammatory and anti-obesity properties. However, the effects of TCEE and its underlying mechanisms in the activation of endothelial nitric oxide synthase (eNOS) have not yet been investigated. Increasing the endothelium-derived nitric oxide (NO) production has been known to be beneficial against the development of cardiovascular diseases. In this study, we investigated the effect of TCEE on eNOS activation and NO-related endothelial function and inflammation by using an in vitro system. In endothelial cells (ECs), TCEE increased NO production in a concentration-dependent manner without affecting the expression of eNOS. In addition, TCEE increased the phosphorylation of eNOS at serine 635 residue (Ser635) and Ser1179, Akt at Ser473, calmodulin kinase II (CaMKII) at threonine residue 286 (Thr286), and AMP-activated protein kinase (AMPK) at Thr172. Moreover, TCEE-induced NO production, and EC proliferation, migration, and tube formation were diminished by pretreatment with LY294002 (an Akt inhibitor), KN62 (a CaMKII inhibitor), and compound C (an AMPK inhibitor). Additionally, TCEE attenuated the tumor necrosis factor-α-induced inflammatory response as evidenced by the expression of adhesion molecules in ECs and monocyte adhesion onto ECs. These inflammatory effects of TCEE were abolished by L-NG-nitroarginine methyl ester (an NOS inhibitor). Moreover, chronic treatment with TCEE attenuated hyperlipidemia, systemic and aortic inflammatory response, and the atherosclerotic lesions in apolipoprotein E-deficient mice. Collectively, our findings suggest that TCEE may confer protection from atherosclerosis by preventing endothelial dysfunction.

## 1. Introduction

The endothelium is a monolayered continuous cell sheet lining the luminal surface of vessel walls that not only serves as the cross-bridge of communication between the blood and cells but also actively regulates the functions of surrounding cells through complex signaling pathways [1,2]. Under certain circumstances, such as hypercholesterolemia and atherosclerosis, modified LDL impairs the function of the endothelial nitric oxide synthase (eNOS)/nitric oxide (NO) system and then upregulates the expression of intercellular adhesion molecule-1 (ICAM-1) and vascular cell adhesion molecule-1 (VCAM-1) in endothelial cell (ECs), leading to the recruitment of monocytes into the subendothelial space of the vessel wall [3,4]. These events have been considered as one of the earliest pathophysiological manifestations of atherosclerosis [5,6]. Several lines of evidence have clearly indicated the crucial role of endothelium-derived NO, whereby it regulates various physiological functions, including vessel relaxation, proliferation and migration of ECs, inhibition of platelet activation, and attenuation of inflammatory responses in the vessel wall [7]. Impaired NO production has been considered one of the earliest pathophysiological manifestations for endothelial dysfunction and is highly associated with the prevalence of inflammatory diseases and cardiovascular diseases [8,9,10]. The regulation of eNOS is tightly regulated not only at the transcriptional level but also by post-translational mechanisms [11,12]. It is well known that eNOS can be activated by several physical and chemical stimuli, such as shear stress, estrogen, or bradykinin, and through kinase-dependent signaling pathways, including the PI3K/Akt, calmodulin kinase II (CaMK II), or AMP-activated protein kinase (AMPK) pathways [11,12]. Increasing the activity of eNOS-NO signaling has been considered a therapeutic strategy for the treatment of cardiovascular diseases [13,14]. 

During the past decade, considerable efforts have been taken to discover the potential of traditional Chinese medicine in many fields, particularly cancers, inflammatory diseases, and cardiovascular diseases [15,16,17,18,19,20]. *Torenia concolor* Lindley var. *formosana* Yamazaki (TC), a plant native to Taiwan that belongs to the Scrophulariaceae family, has been used in traditional Chinese medicine to treat human diseases, including hypertension, stomatitis, hepatitis, pneumonia, and gastroenteritis, etc., owing to its detoxification, anti-inflammatory, and diuretic effects [21,22]. Experimentally, TC extracts have been reported to have excellent anti-inflammatory effects on macrophages and inhibitory effects on lipid deregulation in adipocytes [21,22]. Moreover, the components of TC, such as botulin, betulinic acid, and oleanolic acid, are reported to exert anti-inflammatory, anti-cancer, or anti-hyperglycemic activities [23,24,25]. Although the protective effects of TC extracts on inflammatory diseases have been examined extensively in in vitro and in vivo models [21,22,24]; there is little information about the role of TC extracts in endothelial dysfunction and related cardiovascular diseases. Further investigations of the vascular protective effects of TCEE and its underlying molecular mechanisms in eNOS/NO signaling and EC function are warranted. 

Given the impact of TCEE on inflammatory and metabolic diseases, the present study aimed to characterize the effects of TCEE on ECs and the molecular mechanisms underlying these effects. First, we investigated effects of TCEE and the related molecular mechanisms underlying the activation of eNOS-NO signaling, and then, we ascertained whether TCEE-mediated enhancement of NO bioavailability contributes to its anti-inflammatory property against inflammatory responses. Our third aim was to explore the effects of TCEE on the inflammatory response and the progression of atherosclerosis in apolipoprotein E-deficient (ApoE^−/−^) mice. Our results demonstrated that TECC promotes NO production via the Akt/CaMKII/AMPK/eNOS signaling cascade, which inhibits the inflammatory responses, and ultimately retards the progression of atherosclerosis.

## 2. Results

### 2.1. TCEE Increases NO Production by Increasing the Phosphorylation of eNOS in ECs

To our knowledge, there is no information about the working concentrations of TCEE in ECs. For this reason, we thus tested the effect of TCEE on NO production and physiological function in ECs at a variety of concentrations ranging from 0.125 to 8 μg/mL. To test whether TCEE can activate the eNOS/NO signaling pathway, ECs were treated with the indicated concentrations (0.125, 0.25, 0.5, 1, 2, 4, and 8 μg/mL) of TCEE, and the effects of TCAE on cell viability and NO production were examined. Treating the ECs with 0.125 to 8 μg/mL of TCEE for 24 h did not affect the cell viability (Figure 1A–D) and increased the NO bioavailability (Figure 1E). However, the eNOS protein level was not affected in ECs following the treatment with 0.125 to 8 μg/mL TCEE for 24 h (Figure 1F), suggesting the involvement of increased eNOS activity in the TCEE-induced NO production. Indeed, the increase in NO production was abolished by the pretreatment with L-NAME, a non-selective inhibitor of NOS (Figure 1G), which indicates that the increase in NO bioavailability by TCEE was specifically linked to the function of eNOS. Among the concentrations of TCEE, 2, 4, and 8 μg/mL of TCEE had the most potent effect on NO production; however, there was no statistical significance between these three groups. We thus decided to use 2 μg/mL of TCEE for further experiments. Next, we examined the effect of TCEE on eNOS phosphorylation at various time points. We found that TCEE induced the phosphorylation of eNOS at Ser635 and Ser1179 in a time-dependent manner without affecting its phosphorylation at Ser617 and Thr497 (Figure 2A–D). These results suggest that TCEE increases the NO bioavailability by increasing eNOS phosphorylation. 

### 2.2. Activation of the Akt-CaMKII-AMPK Signaling Pathway Is Essential for the TCEE-Induced Phosphorylation of eNOS and NO Production

The Akt-CaMKII-AMPK signaling pathway is reported to play an important role in eNOS activation and NO production [12]. Next, we examined whether the Akt-CaMKII-AMPK signaling pathway is involved in TCEE-induced eNOS activation and NO production. Our results showed that TCEE time-dependently increased the level of phosphorylation of Akt, CaMKII, and AMPK, which occurred at 15 to 30 min after the treatment (Figure 3A–C). Inhibition of the Akt-CaMKII-AMPK signaling pathway by LY294002 (an Akt inhibitor), KN62 (a CaMKII inhibitor), or compound C (C.C., an AMPK inhibitor) abrogated the TCEE-induced NO production (Figure 3D). By using these pharmacological inhibitors, the up- and downstream relationships of the Akt, CaMKII, and AMPK followed by TCEE treatment were established (Figure 4A–C). Hence, the activation of the Akt-CaMKII-AMPK signaling cascade is required for TCEE-induced eNOS activation and NO production.

### 2.3. The Akt-CaMKII-AMPK Signaling Plays a Key Role in the TCEE-Mediated Promotion of EC Proliferation, Migration, and Tube Formation

Under physiological conditions, increased NO levels can promote EC proliferation, migration, and tube formation, and lead to angiogenesis [26]. We then investigated the effects of TCEE on EC proliferation, migration, and tube formation, and whether the Akt-CaMKII-AMPK/eNOS/NO signaling cascade is involved in the proangiogenic effects of TCEE. Our results revealed that treatment with TCEE enhanced EC proliferation, migration, and tube formation, which were abolished by the pretreatment of the ECs with LY294002, KN62, or C.C. (Figure 5 and Figure 6). These findings suggest that the activation of the Akt-CaMKII-AMPK/eNOS/NO signaling pathway plays a crucial role in the TCEE-mediated benefits associated with the physiological function of ECs.

### 2.4. Activation of eNOS Is Essential for the Anti-Inflammatory Effects of TCEE

TCEE has been reported to have excellent anti-inflammatory effects [21,22]. Based on the above findings, we thus investigated whether eNOS activation mediates the TCEE-conferred anti-inflammatory effects. We demonstrated that tumor necrosis factor-α (TNF-α) increased the expression of ICAM-1 and VCAM-1, as well as monocyte adhesion onto ECs (Figure 7A–E). However, these pro-inflammatory responses elicited by TNF-α were abrogated by the NOS inhibitor L-NAME (Figure 7A–E). These findings suggest the importance of eNOS activation for TCEE to elicit its anti-inflammatory action in ECs.

### 2.5. TCEE Ameliorates Inflammation and Atherosclerosis in ApoE^−/−^ Mice

We next examined the effect of TCEE on inflammation and the progression of atherosclerosis in ApoE^−/−^ mice. As compared with the vehicle treatment, daily TCEE treatment (5 mg/kg) greatly decreased the sizes of atherosclerotic lesions at the aortic sinus, macrophage area, and lipid core in ApoE^−/−^ mice (Figure 8A–C). Treatment with TCEE decreased the serum levels of total cholesterol and non-HDL-c, increased the serum level of HDL-c but did not affect the level of triglycerides in ApoE^−/−^ mice (Figure 8D–G). Furthermore, TCEE decreased the systemic inflammation and aortic inflammation by the evidence that the serum levels of pro-inflammatory cytokines, including tumor necrosis factor-α (TNF-α), interleukin-1β (IL-1β), IL-6, monocyte chemoattractant protein-1 (MCP-1), macrophage inflammatory protein 2 (MIP-2), and the aortic levels of ICAM-1 and VCAM-1, were decreased in TCEE-treated ApoE^−/−^ mice (Figure 9). These results suggest that TCEE has anti-inflammatory and anti-atherogenic actions. 

## 3. Discussion

TCEE is known to have anti-inflammatory, anti-allergic, and lipid-lowering effects [21,22]; however, its biological impact on cardiovascular diseases remains unclear. In this study, we demonstrated that TCEE increased eNOS phosphorylation, NO production, and angiogenic responses in EA.hy926 ECs through the Akt/CaMKII/AMPK signaling pathway. Exposure of ECs to TCEE rapidly increased the phosphorylation of eNOS at Ser635 and Ser1179, and further increased NO production. In addition, the phosphorylation of Akt, CaMKII, and AMPK is involved in TCEE-induced NO production, which is consistent with the results of studies by Ching et al. and Guo et al., who found that Akt, CaMKII, and AMPK are the key kinases in eNOS phosphorylation [26,27]. NO is known to be the most important modulator in regulating the physiological functions of ECs, such as proliferation, migration, and angiogenesis [28,29]. Indeed, our findings showed that TCEE promoted EC proliferation, migration, and angiogenesis in an NO-dependent manner. Moreover, NO has an anti-inflammatory action and thus, inhibits the activation of circulating leukocytes [30]. Moreover, TCEE inhibited the TNFα-induced upregulation of the adhesion molecules ICAM-1 and VCAM-1, as well as the monocyte adhesion onto ECs. Notably, these beneficial effects of TCEE can also be observed in primary human microvascular ECs (Appendix A). Our in vivo findings further support this notion by evidencing that TCEE decreased the systemic and aortic inflammation and thus delayed the progression of atherosclerosis in ApoE^−/−^ mice. Collectively, TCEE may exert its anti-inflammatory and anti-atherogenic effects by preventing EC dysfunction (Figure 10). 

Notably, EC dysfunction is the key event in the early stage of cardiovascular diseases, including hypertension and atherosclerosis [31,32]. It is characterized by decreased NO bioavailability and increased expression of adhesion molecules in atherosclerosis [33,34]. Modulating the activation of eNOS-NO signaling has been suggested as a therapeutic strategy for treating or preventing the progression of cardiovascular diseases [4,35]. For instance, Xing et al. and Li. et al. have reported that supplementation with resveratrol or salidroside improves endothelial dysfunction and alleviates atherosclerosis in ApoE^−/−^ mice [36,37]. Furthermore, treatment with *Morus alba* extract activates eNOS signaling and maintains blood pressure homeostasis in mice [38]. Our previous study has demonstrated that capsaicin and evodiamin, which are the agonists of transient receptor potential vanilloid 1, retard the progression of atherosclerosis by activating the eNOS/NO signaling pathway [13,15,26,39]. In addition, statins, which are lipid-lowering clinical drugs, confer cardiovascular benefits via pleiotropic effects, such as increased NO production and anti-inflammatory activities [8,40]. In this study, our data confirmed this notion by evidencing that TCEE activates the eNOS-NO signaling pathway and attenuates inflammatory responses; however, the effects of TCEE on atherosclerosis and hypertension, as well as the molecular mechanisms underlying these effects, are still unknown. To this end, investigations targeting the key events in the development of atherosclerosis are warranted. 

Importantly, previous studies have reported that 11 bioactive components, including lupeol, stigmasterol, β-sitosterol, betulin, betulinic acid, oleanolic acid, maslinic acid, alphitolic acid, 3-epimaslinic acid, augustic acid, and β-sitosterol-3-0-β-d-glucoside, are found in TC extracts [21,24,25]. Moreover, growing evidence has suggested that these components have protective effects against cardiovascular and inflammatory diseases [11,41]. For example, lupeol is reported to have cardioprotective and anti-inflammatory effects by downregulating the expression of TNF-α, IL-2, IL-4, IL-5, IL-6, IL-11, and prostaglandin E_2_ [11,41]. Impaired metabolism of phytosterols, including β-sitosterol and stigmasterol, causes phytosterolemia, which is closely related to early atherosclerosis [10,42]. Supplementation with phytosterols retards the progression of atherosclerosis by reducing the cholesterol absorption [10,42]. Although these studies have pointed out the beneficial effects of many components of TC extracts, in this study, we cannot ascertain which component accounts for the TCEE-mediated enhancement of NO bioavailability and anti-inflammatory effects in ECs. Therefore, further investigation is needed for studying the protective effects of TCEE components and their regulatory mechanisms in cardiovascular diseases. 

Moreover, the purification methods by which the TC extracts are isolated may affect their bioactivity [21,22,23,24,25]. Previous studies have reported that TC extracts can be purified using water, ethanol, acetate extract, and *n*-butanol [23,24,25]. These TC extracts may have a variety of effects in regulating the physiological functions of the cardiovascular system and the development of cardiovascular diseases [22,24]. For instance, all extracts obtained by various purification methods have been shown to have a similar inhibitory effect on lipopolysaccharide-induced NO production in macrophages [22,24]; however, compared with other TC extracts, the TC extract purified using *n*-butanol has the strongest efficacy in inhibiting lipid accumulation via the activation of peroxisome proliferator-activated receptorγ in adipocytes [22]. However, the effects of other TC extracts, including TCEAE, TCBUE, and TCWE, on the NO-mediated regulation of EC functions and the underlying mechanism require further investigations. Because the hydrophobic property of TCEE facilitates its absorption into cells, compared to the case for hydrophilic extracts, we used TCEE throughout our study. Endothelium-derived NO plays an important role in regulating the vascular homeostasis, and has a protective role against the deregulation of lipid metabolism and inflammation, which are the hallmarks of cardiovascular and metabolic diseases [43,44,45]. Our findings further confirmed the beneficial effects of TCEE on vascular biology via the enhancement of NO bioavailability. Collectively, these observations from our study or others suggest that TC extracts may have therapeutic potential for treating inflammatory and metabolic diseases. 

However, our study contains the limitation that we only used the antibodies to detect the phosphorylation sites of eNOS; however, the antibodies may have nonspecific cross-reaction on various target sites. The kinase assay using ^32^P labelling of eNOS could be the more precise method and further support our findings regarding the status of eNOS phosphorylation. Nevertheless, we did not detect eNOS phosphorylation by the kinase assay due to the reason that our labs are not certified to perform experiments using radioactive substance. On the other hand, we did not study the mechanism how TCEE starts the signaling cascade in this study. Whether there is a specific receptor or target protein to TCEE requires further investigations. 

In conclusion, this study demonstrates that TCEE has the beneficial effects on angiogenesis and the inhibition of inflammation, by activating the eNOS-NO signaling pathway, which confers protective effects from atherosclerosis. Our study provides advanced information about the beneficial effects of TCEE in regulating endothelial function, inflammatory responses, and the progression of atherosclerosis. These findings may broaden the biological significance and biomedical implications of TCEE in the treatment of cardiovascular and metabolic diseases.

## 4. Materials and Methods

### 4.1. Reagents

Mouse antibody for α-tubulin, LY294002, Griess reagent, 3-(4,5-Dimethylthiazol-2-yl)-2,5-diphenyltetrazolium bromide (MTT) assay kit, N-nitro-L-arginine methyl ester (L-NAME), and phosphatase inhibitor cocktails 1 and 2 were obtained from Sigma-Aldrich (St. Louis, MO, USA). Compound C was procured from Calbiochem (San Diego, CA, USA). Rabbit antibodies for phosphor-eNOS at Ser617, 635, and 1179, phosphor-eNOS at Thr497, phosphor-Akt at Ser473, phosphor-CaMKII at Thr286, phosphor-AMPK at Thr172, intercellular adhesion molecule-1 (ICAM-1), and vascular cell adhesion molecule-1 (VCAM-1) were purchased from Cell Signaling Technology (Beverly, MA, USA). Rabbit antibodies for eNOS, Akt, CaMKII, and AMPK were obtained from Santa Cruz Biotechnology (Santa Cruz, CA, USA). Matrigel was procured from BD Biosciences (San Jose, CA, USA). ELISA kits for cytokines were obtained from R&D systems (Minneapolis, MN, USA). Cholesterol and triglyceride assay kits were from Randox (Crumlin, County Antrim, UK).

### 4.2. Cell Culture

EA.hy926 ECs and THP-1 cells were obtained from the ATCC (Manassas, VA, USA) and cultured in DMEM or RPMI 1640 medium supplemented with 10% fetal bovine serum (FBS), 100 U/mL penicillin, and 100 μg/mL streptomycin (HyClone, Logan, UT, USA). EA.hy926 cells were grown until 100% confluence and then used for further experiments. Human microvascular ECs (HMECs) were obtained from the Centers for Disease Control (Atlanta, GA, USA) and grown in Medium 200 (Cascade Biologics Inc., Portland, OR, USA) supplemented with low serum growth supplement containing 2% FBS, 1 μg/mL hydrocortisone, 10 ng/mL human epidermal growth factor, 3 ng/mL human fibroblast growth factor, and 10 μg/mL heparin.

### 4.3. TC Extraction

The TC was crushed to a powder and then was extracted with 95% ethanol followed by partition extraction by different solvents. For the ethanol extraction, TC powder was first soaked in 95% ethanol for 3 days and then the ethanol was removed by a vacuum pump to produce analytes at concentrate form. The concentrated crude extract was weighed and degassed by the nitrogen gas bubbling to remove the residual ethanol. For partition extraction, the aliquot of the crude extract was re-dissolved with ddH_2_O and placed into a partition-extraction bottle, and ethyl acetate and *n*-butanol were sequentially added, in the order of increasing solvent polarity. The two resulting organic extracts were referred to as TCEAE, composed of TC and ethyl acetate, and TCBUE, composed of TC and *n*-butanol. The concentrated aqueous phase remaining after the extraction was referred to as TCWE, which was composed of TC and water. The ethanol-free, concentrated extract was not used in the partition extraction, which was referred to as TCEE. The organic extracts TCEAE, TCBUE, and TCEE were re-dissolved in absolute ethanol and the TCWE extract was mixed with PBS to generate the stock solutions. All the stock solutions were filtered to remove endotoxin by ELGA LabWater Biofilter and stored at −20 °C until further use. In this study, we only examined the effect of TCEE on the physiological functions of ECs.

### 4.4. Cell Viability Assay

MTT assay, cell counting kit-8 (CCK-8) assay, and lactate dehydrogenase (LDH) assay were performed according to the manufacturer’s instructions for evaluating the cell viability. First, the cells were incubated with or without indicated concentrations of TCEE for 24 h; MTT reagent was then added to the cells, followed by incubation for 3 h. On the other hand, the cultured medium was mixed with corresponding reagents of CCK-8 assay and LDH assay for 30 min. The absorbance of these samples was measured at 570 (MTT assay) and 490 nm (CCK-8 and LDH assays), respectively. The cell viability was normalized relative to the cellular protein concentration. Cells incubated with the vehicle were considered 100% viable. 

### 4.5. Protein Extraction and Western Blot Analysis

Cells were lysed with the SDS lysis buffer, which contained 1% Triton, 0.1% SDS, 0.2% sodium azide, 0.5% sodium deoxycholate, and proteinase inhibitors (1 mmol/L PMSF, 10 mg/mL aprotinin, and 1 mg/mL leupeptin). The lysates were centrifuged at 12,000 rpm for 5 min and the resulting supernatants were collected. The extracted protein was quantified by protein assay. The proteins were separated by SDS-PAGE; the protein bands were transferred onto BioTrace PVDF membranes. After being blocked with 5% skim milk, the blots were incubated with the primary antibodies, and then with the secondary antibodies. The protein bands were detected by using an enhanced chemiluminescence kit and the level of protein expression was quantified by using ImageQuant 5.2 (Healthcare biosciences, Philadelphia, PA, USA).

### 4.6. Determination of Nitrite Production

Nitric oxide (NO) has an extremely short half-life and is quickly metabolized into nitrite and nitrate. Thus, the level of nitrite was determined by the Griess assay to assess NO production. The cell culture medium was mixed with an equal volume of Griess reagent, and then azo dye production was determined by measuring the absorbance of these samples at 540 nm after 15 min of incubation at room temperature. The level of nitrite was normalized relative to the cellular protein concentration. Sodium nitrite was used as a standard.

### 4.7. Cell Proliferation Assay

EA.hy926 cells were cultured on 12-well plates in DMEM containing 10% FBS and subjected to serum starvation for 12 h. After treatment, the cells were labeled BrdU with for 4 h. BrdU incorporation was measured at least in triplicate at each time by the use of a cell proliferation ELISA colorimetric kit (Roche, Mannheium, Germany). The absorbance was measured by spectrophotometry at 370 nm and referred at 492 nm.

### 4.8. Cell Migration Assay

The transwell migration assay was performed using modified chambers inserted into 24-well plates. First, 5 × 10^6^ cells suspended in 300 μL of DMEM were added into the upper chamber. On the other hand, 500 μL of DMEM was added into the lower chamber. After 18 h of incubation, the medium and unmigrated cells in the upper chamber were removed, and the migrated cells in the lower side of the membranes were stained with crystal violet. Images were digitally captured under a Nikon TE2000-U florescence microscope with an image analysis system (QCapture Pro 6.0, QImaging, Surrey, BC, Canada). 

### 4.9. Matrigel Angiogenesis Assay

Matrigel was coated onto 3.5 cm dishes and allowed to polymerize for 30 min at 37 °C. ECs were seeded onto the Matrigel layer and subjected to the indicated treatments for 9 h. The tube formation ability of the ECs was quantified by counting the number of branch points and the images were photomicrographed under a Nikon TE2000-U florescence microscope with an image analysis system.

### 4.10. In Vitro Mononuclear-Endothelial Cell Adhesion Assay

The adherence of THP-1 cells onto the ECs was examined under static conditions. For fluorescence staining, THP-1 cells were incubated with BCECF-AM at 37 °C for 1 h and then added into the culture medium at a concentration of 1 × 10^5^ cells/mL; the PHP-1 cells were incubated with the ECs for 1 h. Fluorescence of the cell lysates was measured by fluorometry (Molecular Devices, Sunnyvale, CA, USA) with excitation and emission wavelengths of 485 and 525 nm, respectively. Images were photomicrographed under a Nikon TE2000-U microscope with an image analysis system.

### 4.11. Serum Lipid Profile Analysis

The serum levels of total cholesterol, high-density lipoprotein cholesterol (HDL-c), non-HDL-c, and triglycerides were measured by use of the blood biochemical analyzer Spotchem EZ SP 4430 (ARKRAY, Inc., Kyoto, Japan) or assay kits.

### 4.12. Measurement of Inflammatory Cytokines

The serum concentrations of pro-inflammatory cytokines, including TNF-α, IL-1β, MCP-1, IL-6, and MIP-2, were measured by use of ELISA kits.

### 4.13. Mice

All animal experiments were approved by the Animal Care and Utilization Committee of National Yang-Ming University and the investigation conformed to the Guide for the Care and Use of Laboratory Animals (Institute of Laboratory Animal Resources, eighth edition, 2011). ApoE^−/−^ mice were purchased from Jackson Laboratory (Bar Harbor, Maine). ApoE^−/−^ mice at 16 weeks old received oral daily treatment with TCEE (5 mg/kg body weight) or vehicle by gastric gavage for 4 weeks. At the end of the experiment, mice were euthanized with CO_2_ and the hearts were isolated and fixed with 4% paraformaldehyde and then embedded in paraffin. The deparaffinized sections were subjected to H&E staining and the quantification of atherosclerotic lesions at the aortic sinus was performed. The lysates of atherosclerotic aortas were subjected to Western blot analysis to evaluate the expression of proteins.

### 4.14. Statistical Analysis

Results are presented as the mean ± SEM from five independent experiments or 10 mice. Mann–Whitney test was used to compare two independent groups. Kruska–Wallis test, followed by the Bonferroni post hoc analyses, were used for comparing data from multiple groups. SPSS software v8.0 (SPSS Inc. Chicago, IL, USA) was used to perform the statistical analyses. *p* < 0.05 was considered statistically significant.

## Figures and Tables

**Figure 1 ijms-21-01532-f001:**
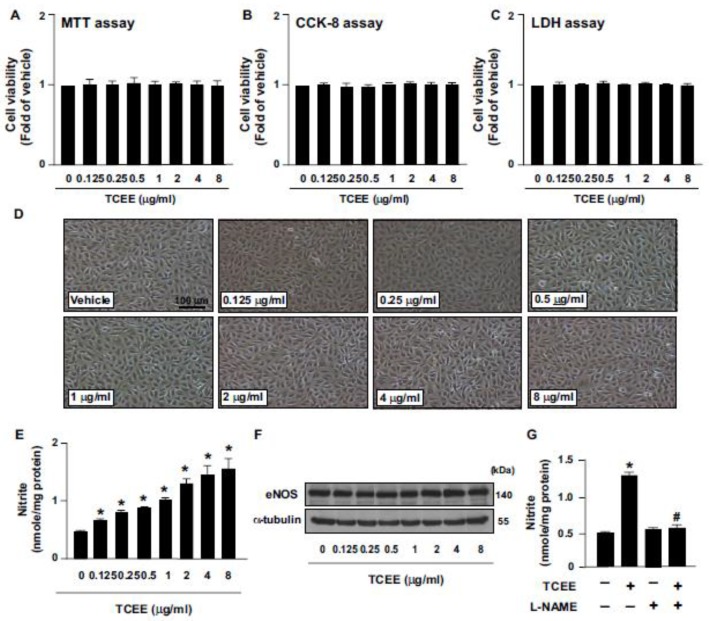
Effects of *Torenia concolor* Lindley var. *Formosama* Yamazaki ethanol extracts (TCEE) on cell viability and NO bioavailability in ECs. ECs were treated with the indicated concentrations (0.125-8 μg/mL) of TCEE for 24 h. (**A**–**C**) The cell viability by MTT assay, CCK-8 assay, and LDH assay. (**D**) Microscopic analysis of cells. (**E**) The level of nitrite in culture medium by Griess assay. (**F**) Western blot analysis of total eNOS and α-tubulin. (**G**) ECs were pretreated with L-NAME (400 μM) for 2 h, and then with TCEE (2 μM) for 24 h. The level of nitrite in the culture medium was measured. Data are mean ± SEM from 5 independent experiments. * *p* < 0.05 vs. vehicle-treated group; ^#^
*p* < 0.05 vs. TCEE-treated group.

**Figure 2 ijms-21-01532-f002:**
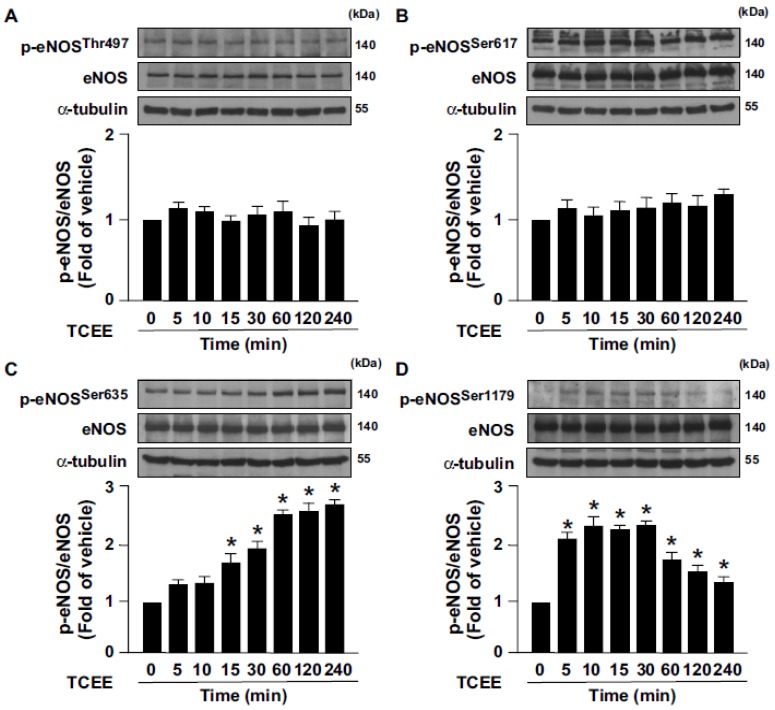
TCEE induces endothelial NO synthase (eNOS) activation in ECs. ECs were treated with TCEE (2 μg/mL) for the indicated times. (**A**–**D**) Cells were lysed and subjected to Western blot analysis for quantifying the levels of phosphorylated eNOS (at Thr495, Ser617, Ser635, and Ser1177) or total eNOS protein. Data are mean ± SEM from 5 independent experiments. * *p* < 0.05 vs. vehicle-treated group.

**Figure 3 ijms-21-01532-f003:**
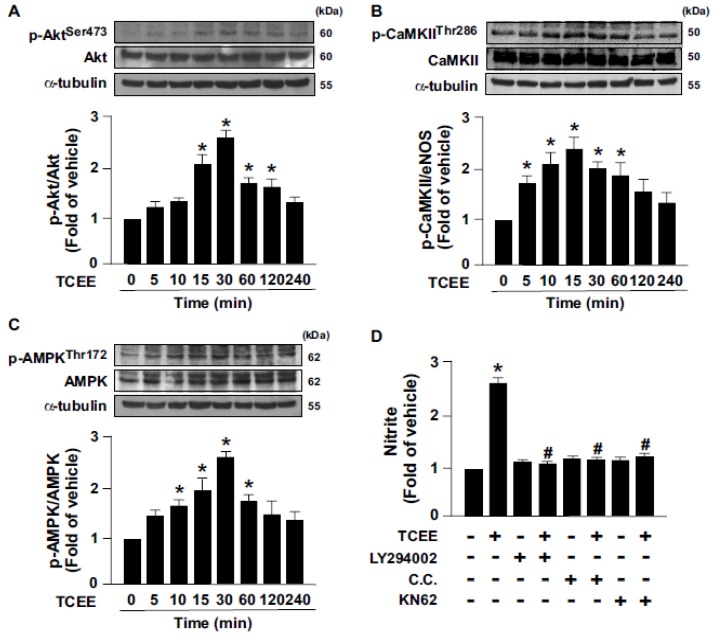
Akt-CaMKII-AMPK signaling is involved in the TCEE-mediated enhancement of NO bioavailability in ECs. (**A**–**C**) ECs were treated with TCEE (2 μg/mL) for the indicated times. Western blot analysis of phosphorylated or total of Akt, CaMKII, and AMPK and α-tubulin. (**D**) ECs were pretreated with LY294002 (10 μM), compound C (C.C., 10 μM), or KN62 (10 μM) for 2 h, and then with TCEE (2 μg/mL) for 24 h. Griess assay of the nitrite level in the culture medium. Data are mean ± SEM from 5 independent experiments. * *p* < 0.05 vs. vehicle-treated group; ^#^
*p* < 0.05 vs. TCEE-treated group.

**Figure 4 ijms-21-01532-f004:**
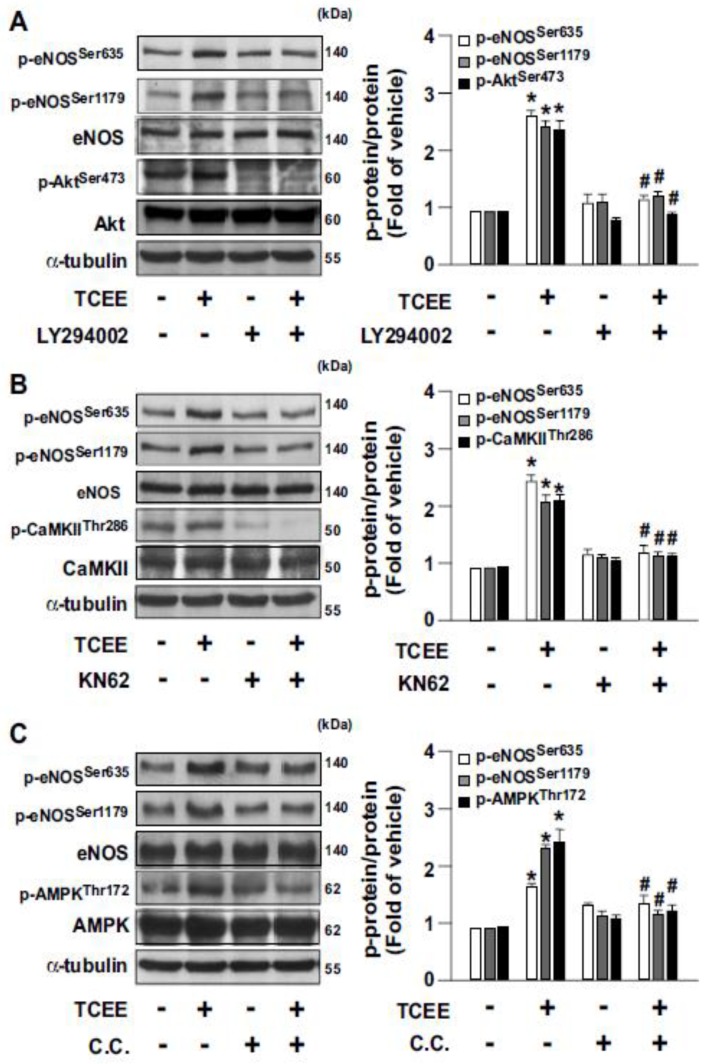
The up- and downstream relationships of the Akt, CaMKII, and AMPK signaling cascade in TCEE-treated ECs. (**A**–**C**) ECs were pretreated with LY294002 (10 μM), compound C (C.C., 10 μM), or KN62 (10 μM) for 2 h, and then with TCEE (2 μg/mL) for 30 min. Western blot analysis of phosphorylated or total of Akt, CaMKII, AMPK, eNOS, and α-tubulin. Data are mean ± SEM from 5 independent experiments. * *p* < 0.05 vs. vehicle-treated group; ^#^
*p* < 0.05 vs. TCEE-treated group.

**Figure 5 ijms-21-01532-f005:**
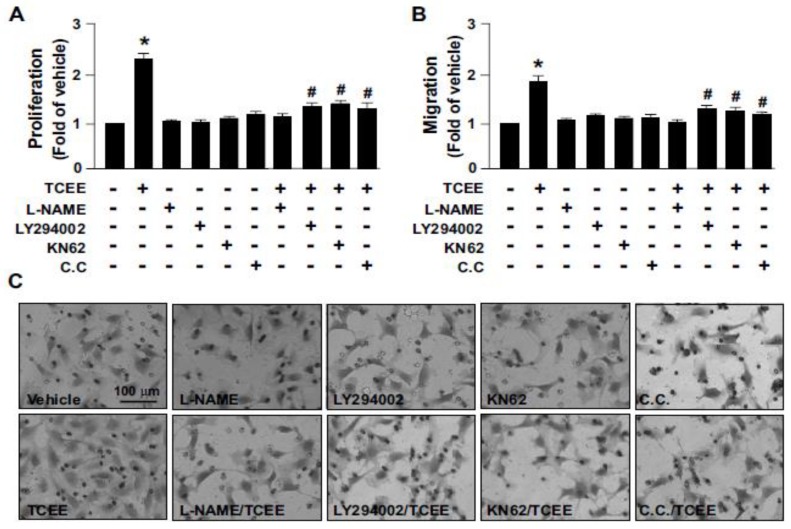
The signaling cascade involved in the TCEE-induced promotion of EC proliferation and migration. ECs were pretreated with LY294002 (10 μM), compound C (C.C., 10 μM), or KN62 (10 μM) for 2 h, then with TCEE (2 μg/mL) for 8 h. (**A**) MTT assay of EC proliferation. (**B**,**C**) EC migration was evaluated by the trans-well assay after TCEE treatment. The number of migrating cells was counted compared to the case for the vehicle-treated cells, and the images were generated by microscopy. Data are mean ± SEM from 5 independent experiments. * *p* < 0.05 vs. vehicle-treated group; ^#^
*p* < 0.05 vs. TCEE-treated group.

**Figure 6 ijms-21-01532-f006:**
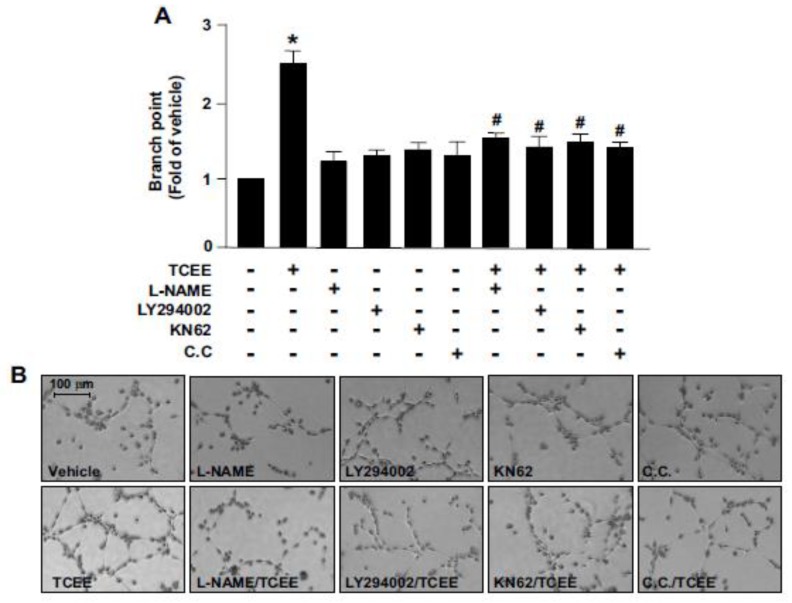
The signaling cascade involved in TCEE-induced promotion of EC tube formation. ECs were pretreated with LY294002 (10 μM), compound C (C.C., 10 μM), or KN62 (10 μM) for 2 h or l-NAME (400 μM) for 1 h, then with TCEE (2 μg/mL) for 8 h. (**A**,**B**) ECs were cultured in the attachment matrix in the indicated treatments. Tube formation was visualized; the bar graphs indicate the branch points as a fold of those in case of the vehicle-treated control in five randomly selected microscopic fields. Data are mean ± SEM from 5 independent experiments. * *p* < 0.05 vs. vehicle-treated group; ^#^
*p* < 0.05 vs. TCEE-treated group.

**Figure 7 ijms-21-01532-f007:**
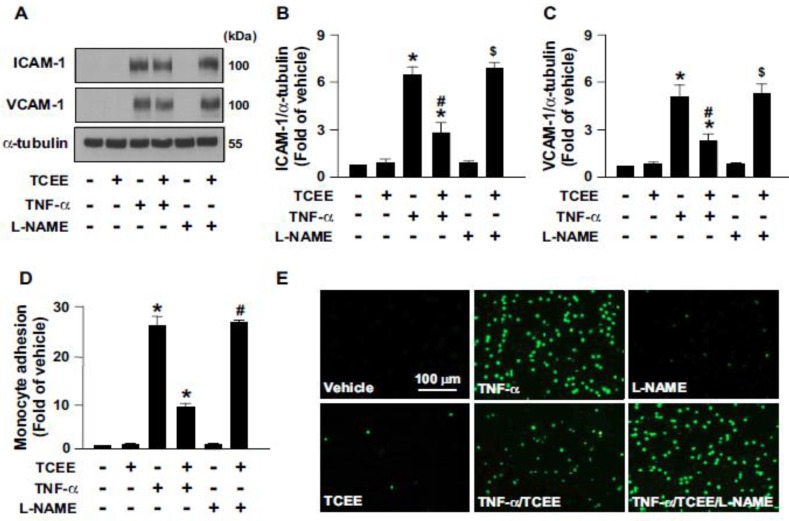
TCEE inhibits the TNFα-induced adhesion of monocytes onto ECs and the protein expression of adhesion molecules in ECs. ECs were pretreated with TNFα (10 ng/mL) or the vehicle for 12 h, followed by treatment with L-NAME (400 μM) for 1 h and/or TCEE (2 μg/mL) for another 18 h. (**A**–**C**) Cellular lysates were subjected to Western blot analysis for quantifying the protein expression of ICAM-1, VCAM-1, and α-tubulin. (**D**,**E**) BCECF-AM-labeled human monocyte THP-1 cells (1 × 10^5^) were added and incubation with ECs for 1 h. The cellular lysates were subjected to fluorometry and photo-micrographed. Data are mean ± SEM from 5 independent experiments. * *p* < 0.05 vs. vehicle group; ^#^
*p* < 0.05 vs. TCEE-treated group; ^$^
*p* < 0.05 vs. TNF-α +TCEE-treated group.

**Figure 8 ijms-21-01532-f008:**
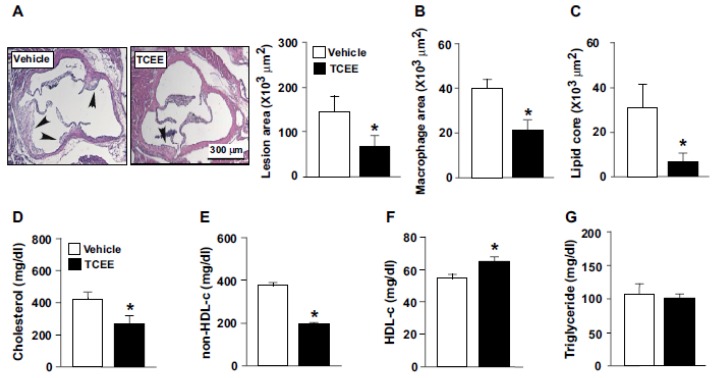
TCEE ameliorates hyperlipidemia and atherosclerosis in ApoE^−/−^ mice. Male ApoE^−/−^ mice at age 16 weeks were orally administered with TCEE (5 mg/kg/day) or vehicle for 4 weeks. (**A**) Atherosclerotic lesions at aortic roots were stained with H&E. (**B**) The area of macrophages in the atherosclerotic lesions. (**C**–**G**) Serum levels of total cholesterol, non-high-density lipoprotein cholesterol (non-HDL-c), HDL cholesterol (HDL-c), and triglycerides. Data are mean ± SEM from 10 mice. * *p* < 0.05 vs. vehicle. Scale bar = 300 μm.

**Figure 9 ijms-21-01532-f009:**
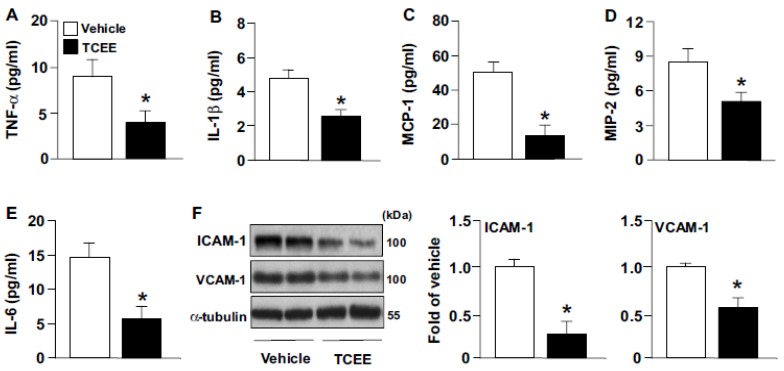
TCEE attenuates systemic and aortic inflammation in ApoE^−/−^ mice. Male ApoE^−/−^ mice at age 16 weeks were orally administered with TCEE (5 mg/kg/day) or vehicle (oil) for 4 weeks. (**A**–**E**) ELISA of serum levels of TNF-α, IL-1β, MCP-1, MIP-2 and IL-6. (**F**) Western blot analysis of protein levels of ICAM-1, VCAM-1, and -α tubulin in aortas from ApoE^−/−^ mice. Data are mean ± SEM from 10 mice. * *p* < 0.05 vs. vehicle.

**Figure 10 ijms-21-01532-f010:**
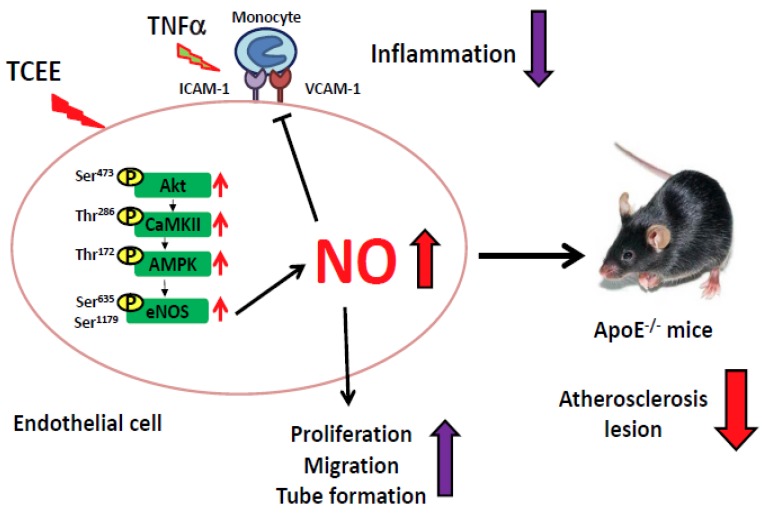
Schematic illustration of the proposed mechanism underlying the TCEE-mediated anti-inflammatory effect in ECs and atherosclerosis. As shown, challenge with TCEE elicits the activation of the Akt/CaMKII/AMPK/eNOS/NO pathway, which in turn promotes pro-angiogenic activity and inhibits the pro-inflammatory response by TNF-α in ECs and retards the atherosclerosis in ApoE^−/−^ mice.

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
