# Peer review of "Endothelial Nitric Oxide Mediates the Anti-Atherosclerotic Action of Torenia concolor Lindley var. Formosama Yamazaki"

_ijms, 2020, doi:10.3390/ijms21041532_

Round 1

Reviewer 1 Report

This is the second time that I have reviewed this manuscript. I have read the comments from the authors from my initial review and I am now satisfied they have addressed the areas which I previously had expressed concerns.

Re: [IJMS] Manuscript ID: ijms-697427 - Review Request-Resubmission of ijms-584125

I would ask that the authors acknowledge in the discussion that there is  a limitation of relying on antibodies to detect phosphorylation and ask that they add a comment that 32P labelling of eNOS could further support their findings.

I thank them for their thorough responses and would be happy for the manuscript to be published including the new figures.

Author Response

I would ask that the authors acknowledge in the discussion that there is a limitation of relying on antibodies to detect phosphorylation and ask that they add a comment that 32P labelling of eNOS could further support their findings.

Response: We thank the reviewer for the professional suggestion. In response to the reviewer’s suggestion, we have addressed this point as our limitation in the discussion of the revised manuscript. Now the paragraph read as “However, our study contains the limitation that we only used the antibodies to detect the phosphorylation sites of eNOS; however, the antibodies may have nonspecific cross-reaction on various target sites. The kinase assay using 32P labelling of eNOS could be the more precise method and further support our findings regarding with the status of eNOS phosphorylation. Nevertheless, we did not detect eNOS phosphorylation by the kinase assay due to the reason that our labs are not certified to perform experiments using radioactive substance.”

We sincerely hope that the reviewer can approve our viewpoint.

Reviewer 2 Report

The study by Li-Ching Cheng et al. investigated the effects of the Torenia Concolor Lindley var. formosama Yamazaki extract (TCEE) on the development of atherosclerosis in the apo E knockout mouse. The molecular mechanisms were studied in the endothelial cells and were focused on the regulation of endothelial nitric oxide synthase (eNOS).

The authors demonstrated in vitro that TCEE activated the eNOS-NO signaling pathway and attenuated inflammatory responses. In animal model, TCEE attenuated hyperlipidemia, inflammation, and decreased the atherosclerotic lesions in apolipoprotein E-deficient mice.

Comments:

The manuscript consists of two parts, in vitro and in vivo. Describing of in vitro experiments is too long. Some Figures could be removed, for example cell viability etc., in which no regulation has been found. Molecular size of the proteins in Western Blot analysis should be added to Figures. Is it something known about TCEE induced mechanism that starts the signaling cascade in the cell (receptor etc.)? Which part of the plant was used for  drug extraction? What is known about the pharmacokinetics of TCEE?

Author Response

The manuscript consists of two parts, in vitro and in vivo. Describing of in vitro experiments is too long. Some Figures could be removed, for example cell viability etc., in which no regulation has been found. Molecular size of the proteins in Western Blot analysis should be added to Figures. Is it something known about TCEE induced mechanism that starts the signaling cascade in the cell (receptor etc.)? Which part of the plant was used for drug extraction? What is known about the pharmacokinetics of TCEE? 

Response: We thank the reviewer for allowing us to explain more about these important issues. Because the results of cell viability by MTT assay, CCK-8 assay and LDH assay were requested by other reviewers; therefore, we decided to keep these data in our manuscript. We sincerely hope that the reviewer could approve our arrangement for these data.

In response to the reviewer’s suggestion, the molecular size of the proteins in western blot analysis had been added to figures of the revised manuscript. In this study, the whole plant of TC was used for drug extraction. To our knowledge, little is known about how TCEE activates the cellular signaling cascade and its pharmacokinetics of TCEE in vitro and in vivo. We sincerely hope that the reviewer could approve our viewpoint.

Reviewer 3 Report

In this paper Cheng and co-workers reported the anti-inflammatory and anti-obesity properties of Torenia concolor Lindley var. formosana Yamazaki; in particular they studied the role of eNOS in the extracts properties.

The paper is well written and the authors demonstrated clearly the role of eNOS activation in the beneficial effects of Torenia concolor Lindley var. formosana Yamazaki in the development of cardiovascular diseases.

Some points should be discuss better by the authors before the acceptance of the manuscript.

In the experimental section the authors described the extraction method of Torenia concolor Lindley var. formosana Yamazaki, in particular they obtained 3 organic extracts (TCEAE, TCBUE, and TCEE) and one water extract (TCWE). In the discussion the authors wrote “all the TC extracts mentioned above have similar efficacies on eNOS-mediated NO production (data not shown)”, but through the text seems that they tested only the TCEE extract. The authors underline that many bioactive components should be present in the extracts, and that the aim of their work is to point out the beneficial effects of the extracts not of a single component. In any case, first of all, I think that the authors should clarify in the text, if they used only the TCEE extract or all the extracts obtained. Secondly I think that the different extracts contain different bioactive compounds, the extracts were obtained with different solvent that have different extraction power, so probably not all the bioactive compounds should be present in all the extracts and not in the same quantity, and this difference should be important for their activity.

In the middle of the introduction the authors introduced TCEE (“However, whether TCEE has such beneficial effects on eNOS activation and EC functions, and the molecular mechanisms underlying these effects remain unclear”), but in this way it’s not clear what TCEE is and why they introduced this concept, the authors should report here some data of previously study on Torenia concolor Lindley var. formosana Yamazaki.

The authors should also explain why they used determined concentrations of extracts, these data derived from previous studies?

The manuscript should be accepted after minor revision in the text.

Author Response

In this paper Cheng and co-workers reported the anti-inflammatory and anti-obesity properties of Torenia concolor Lindley var. formosana Yamazaki; in particular they studied the role of eNOS in the extracts properties.

The paper is well written and the authors demonstrated clearly the role of eNOS activation in the beneficial effects of Torenia concolor Lindley var. formosana Yamazaki in the development of cardiovascular diseases.

Some points should be discussed better by the authors before the acceptance of the manuscript.

1. In the experimental section the authors described the extraction method of Torenia concolor Lindley var. formosana Yamazaki, in particular they obtained 3 organic extracts (TCEAE, TCBUE, and TCEE) and one water extract (TCWE). In the discussion the authors wrote “all the TC extracts mentioned above have similar efficacies on eNOS-mediated NO production (data not shown)”, but through the text seems that they tested only the TCEE extract. The authors underline that many bioactive components should be present in the extracts, and that the aim of their work is to point out the beneficial effects of the extracts not of a single component. In any case, first of all, I think that the authors should clarify in the text, if they used only the TCEE extract or all the extracts obtained. Secondly I think that the different extracts contain different bioactive compounds, the extracts were obtained with different solvent that have different extraction power, so probably not all the bioactive compounds should be present in all the extracts and not in the same quantity, and this difference should be important for their activity.

Response: We fully agree with the reviewer’s viewpoint. Indeed, we examined the effects of TCEE and TCWE on eNOS-mediated NO production. However, only TCEE was used for the mechanistic study, functional assays and in vivo study. In response to the reviewer’s suggestion, we have deleted the description about “all the TC extracts mentioned above have similar efficacies on eNOS-mediated NO production (data not shown)” and added the description “In this study, we only examined the effect of TCEE on physiological functions of ECs” in the experimental section and the description “However, the effects of other TC extracts including TCEAE, TCBUE and TCWE on the NO-mediated regulation of EC functions and the underlying mechanism remain further investigations.” in the discussion section of the revised manuscript.

We sincerely hope that the reviewer can approve our revision.

2. In the middle of the introduction the authors introduced TCEE (“However, whether TCEE has such beneficial effects on eNOS activation and EC functions, and the molecular mechanisms underlying these effects remain unclear”), but in this way it’s not clear what TCEE is and why they introduced this concept, the authors should report here some data of previously study on Torenia concolor Lindley var. formosana Yamazaki.

Response: We thank the reviewer for reminding us this important issue. We apologize for this negligence in scientific writing; this is a redundant sentence in introduction section. In response to the reviewer’s suggestion, we have deleted this sentence in our revised manuscript.

3. The authors should also explain why they used determined concentrations of extracts, these data derived from previous studies?

Response: We thank for the reviewer allowing us to explain more about this important issue. To our knowledge, there is no information about the working concentrations of TCEE in endothelial cells. For this reason, we thus test the effect of TCEE on NO production and physiological function in endothelial cells at a variety of concentrations (range from 0.125~8 μg/ml). Our data showed that all these concentrations of TCEE could significantly induce NO production without inducing cytotoxicity (Figure 1). Among these concentrations of TCEE, 2, 4 and 8 μg/ml of TCEE had most potent effect on NO production; however, there was no statistical significance between these three groups. We thus decided to use 2 μg/ml of TCEE for further experiments. In response to the reviewer’s suggestion, we have added the description about the concentrations of TCEE in the section 2.1 of results of the revised manuscript.

We sincerely hope that the reviewer can approve our viewpoint.

4. The manuscript should be accepted after minor revision in the text.

Response: We fully agree with the reviewer’s viewpoint. We have revised our manuscript according to the reviewer’s suggestion.